# 6-Shogaol as a Novel Thioredoxin Reductase Inhibitor Induces Oxidative-Stress-Mediated Apoptosis in HeLa Cells

**DOI:** 10.3390/ijms24054966

**Published:** 2023-03-04

**Authors:** Shoujiao Peng, Shaopeng Yu, Junmin Zhang, Jiange Zhang

**Affiliations:** 1The Research Center of Chiral Drugs, Innovation Research Institute of Traditional Chinese Medicine (IRI), Shanghai University of Traditional Chinese Medicine, Shanghai 201203, China; 2School of Pharmacy, Lanzhou University, Lanzhou 730000, China; 3Shanghai Frontiers Science Center for Traditional Chinese Medicine Chemical Biology, Shanghai University of Traditional Chinese Medicine, Shanghai 201203, China

**Keywords:** natural products, 6-shogaol, thioredoxin reductase, oxidative stress, apoptosis

## Abstract

Inhibition of thioredoxin reductase (TrxR) is a crucial strategy for the discovery of antineoplastic drugs. 6-Shogaol (6-S), a primary bioactive compound in ginger, has high anticancer activity. However, its potential mechanism of action has not been thoroughly investigated. In this study, we demonstrated for the first time that 6-S, a novel TrxR inhibitor, promoted oxidative-stress-mediated apoptosis in HeLa cells. The other two constituents of ginger, 6-gingerol (6-G) and 6-dehydrogingerduone (6-DG), have a similar structure to 6-S but fail to kill HeLa cells at low concentrations. 6-Shogaol specifically inhibits purified TrxR1 activity by targeting selenocysteine residues. It also induced apoptosis and was more cytotoxic to HeLa cells than normal cells. The molecular mechanism of 6-S-mediated apoptosis involves TrxR inhibition, followed by an outburst of reactive oxygen species (ROS) production. Furthermore, TrxR knockdown enhanced the cytotoxic sensitivity of 6-S cells, highlighting the physiological significance of targeting TrxR by 6-S. Our findings show that targeting TrxR by 6-S reveals a new mechanism underlying the biological activity of 6-S and provides meaningful insights into its action in cancer therapeutics.

## 1. Introduction

Morbidity and mortality statistics suggest that cervical cancer is one of the most prevalent cancers among women worldwide, ranking fourth after breast, colorectal, and lung cancers [1]. Despite significant advancements in detection, prevention, and treatment, cervical cancer still affects the lives of many women, particularly in low-income nations [2]. Hysterectomy and chemotherapy are the most commonly used therapies for women with cervical cancer. Therefore, effective treatment of cervical cancer requires the rapid identification and characterization of chemotherapeutic drugs derived from natural small molecules.

Thioredoxin (Trx), thioredoxin reductase (TrxR), and NADPH comprise the thioredoxin system and are ubiquitous in all cells [3]. This system serves as both a critical regulator of numerous cellular redox signaling pathways and a bridge between the universal reduction of NADPH and diverse biological substances [4]. Reduced Trx regulates different intracellular pathways via a general thiol–disulfide exchange reaction, which is converted into oxidized Trx. TrxR then reduces the disulfide in the active site of oxidized Trx to a dithiol using NADPH as an electron donor [4]. Two major isoforms of Trx/TrxR have been identified in mammalian cells: Trx1/TrxR1 in the cytosol and nucleus and Trx2/TrxR2 in the mitochondria [5]. Although mammalian TrxR1 and TrxR2 are localized in separate regions of the cell, their catalytic processes are remarkably similar. As homodimeric flavin proteins, mammalian TrxRs share an essential yet distinct selenocysteine residue at their C-termini [6]. Numerous clinical observations and laboratory studies have indicated that overactivation/dysfunction of TrxRs is closely associated with the onset and progression of diseases such as cancer and neurodegeneration [7]. Therefore, this selenoenzyme is a promising target for cancer therapies.

Natural products derived from daily diets are garnering increasing attention in cancer treatment. The rhizome of ginger, *Zingiber officinale*, has been used for thousands of years in traditional Indian and Southeast Asian medicine and spices to provide a unique flavor [8]. To date, many pharmacological studies have focused on identifying ginger rhizome constituents and exploring their bioactivity [9]. 6-Shogaol (6-S), a naturally occurring compound isolated from dehydrated ginger, has been investigated for various pharmacological effects, including anticancer [10], antioxidative [11], anti-inflammatory [12], anti-allergic [13], and neuroprotective activities [14]. Furthermore, Wu et al. found that 6-S had higher anticancer activity than 6-gingerol and another widely studied phytochemical, curcumin [15]. Growing evidence indicates that various mechanisms accounting for the anticancer activity of 6-S, such as regulating the AKT signaling pathway [16] and p53 signaling pathway [17,18], inducing apoptosis and suppressing autophagy [19], inhibiting STAT3 and NF-κB signaling [20], activating caspase-independent paraptosis [21], and generating reactive oxygen species (ROS), are essential for the cellular action of 6-S [10,17,22,23,24]. However, the mechanism by which 6-S induces ROS production remains poorly defined, and 6-S’s primary cellular target of action remains a matter of contention.

In this study, we showed that 6-S exhibited the most potent cytotoxicity toward HeLa cells compared with 6-gingerol (6-G) and 6-dehydrogingerduone (6-DG). 6-Shogaol inhibited the physiological function of TrxR, leading to the accumulation of ROS and the breakdown of the intracellular redox balance. As a result, 6-S caused oxidative stress in HeLa cells, ultimately leading to cell death via apoptosis. Furthermore, TrxR1 knockdown increased the cytotoxicity of 6-S, suggesting the physiological importance of targeting TrxR with 6-S. Thus, inhibition of TrxR provides a novel mechanism for the biological activity of 6-S and sheds light on the possible application of 6-S in cancer treatment.

## 2. Results

### 2.1. Induction of Cell Death by 6-S

6-Gingerol, 6-dehydrogingerduone, and 6-shogaol, extracted from ginger, have similar structures (Figure 1A). We first investigated the cytotoxic properties of 6-G, 6-DG, and 6-S toward HeLa cells. Among the three compounds, 6-S displayed the most potent cytotoxicity, with an IC50 value of 15.7 μM obtained after 24 h of treatment (Figure 1B). The treatment of HeLa cells with 6-S for different durations revealed that this compound caused a dose- and time-dependent decrease in cell viability (Figure 1C). To assess whether 6-S explicitly targets cancer cells, we employed an Ect1 cell line (an epithelial cell line from the ectocervix) to examine the cytotoxicity of 6-S. Under the same experimental conditions, the cytotoxic effect of 6-S was more pronounced in HeLa cells than in Ect1 cells (Figure 1D), indicating that malignant cells were more susceptible to 6-S than normal cells.

### 2.2. Inhibition of Purified TrxR Activity by 6-S

Analysis of the chemical structure of 6-S revealed the presence of an *α,β*-unsaturated carbonyl, indicating that this compound could potentially inhibit TrxR activity [25,26,27]. The inhibition of 6-S, 6-G, and 6-DG on purified TrxR1 was assessed by an endpoint insulin reduction assay. As shown in Figure 2A, 6-S displayed better inhibition than 6-DG, and 6-S had a weak inhibition effect at 50 µM, so we selected 6-S for the follow-up studies. Furthermore, because TrxR and GR are structurally similar, and Sec residues in TrxR are essential for its biological function, we compared the effect of 6-S on three enzyme preparations: (1) NADPH-reduced recombinant rat TrxR1 (WT TrxR1); (2) U498C TrxR1, in which Sec498 was replaced with a Cys residue; and (3) GR. We observed that 6-S effectively inhibited WT TrxR1 activity but only weakly inhibited U498C TrxR1 activity, while ultimately failing to inhibit the activity of GR (Figure 2B). Thus, 6-S treatment specifically inhibited WT TrxR1 activity by interacting with Sec residues. To supplement the findings shown in Figure 2B, we used molecular docking to evaluate the possible interaction of mammalian TrxR1 with 6-S. As shown in Figure 2C, the α,β-unsaturated carbonyl group of 6-S covalently bonded to TrxR1 at Sec498. The binding pocket of 6-S is composed of two parts, namely Trp407, Leu409, Glu494, Cys497, and Sec498 on Chain A and Ala26, Lys29, Tyr116, Ile347, and Arg351 on Chain B. Four types of bonds formed between 6-S and TrxR1: (1) conventional hydrogen bonds, highlighted in dark green dashed lines (Trp407, Gly499, Sec498, and Tyr116); (2) Carbon-hydrogen bonds, highlighted in light green dashed lines (Leu409, Glu494, and Cys497); (3) alkyl bonds, highlighted in pink dashed lines (Ala26, Lys29, Ile347, and Arg351); (4) covalent bonds, highlighted in purple dashed lines (Sec498) (Figure 2D). We also performed the covalent docking between 6-G/6-DG and mammalian TrxR1, and the results demonstrated that 6-S showed the best docking affinity scores compared with 6-G and 6-DG. According to these findings, 6-S interacts irreversibly with mammalian TrxR1, enabling the formation of covalent bonds between 6-S and TrxR1.

### 2.3. Influence of 6-S on Intracellular TrxR and Trx

As confirmed by the above results, 6-S suppressed pure TrxR activity by targeting Sec498 in TrxR1. To gain further insight into the effect of 6-S in HeLa cells, we investigated the effect of 6-S on cellular TrxR using the fluorescent probe TRFS-green. As shown in Figure 3A, 6-S effectively inhibited the activity of cellular TrxR, and the measurement of the relative fluorescence intensity (R.F.I.) of individual cells is shown in Figure 3B. We also assessed TrxR activity in HeLa cells and Ect1 cells using an endpoint insulin reduction assay. As shown in Figure 3C, treating HeLa cells with 6-S significantly suppressed cellular TrxR activity in HeLa cells, which is consistent with the results shown in Figure 3A, but the cellular TrxR activity in Ect1 cells hardly changed. The dithiol adjacent to the active site of the reduced Trx interacted with a broad range of downstream target proteins via a general thiol–disulfide exchange reaction to form oxidized disulfide (oxidized Trx), which was then reduced back to the reduced thiol form by TrxR by taking electrons from NADPH. As reduced Trx has been identified as an essential cellular enzyme that performs various biological functions, we monitored the redox state of Trx after the 6-S treatment. As shown in Figure 3D,E, treatment with 6-S shifted the Trx redox state to the oxidized form, indicating that TrxR inhibition strongly affected disulfide reduction in Trx. Due to the inhibition of TrxR activity in HeLa cells, many oxidative Trx molecules could not be converted to reduced Trx via TrxR catalysis. As a result, oxidative Trx may eventually be expelled from the cell. These results suggested that the 6-S treatment perturbed the Trx system via inhibiting TrxR in HeLa cells.

### 2.4. Promotion of Oxidative Stress by 6-S

Since TrxR is tightly associated with ROS scavenging, we investigated the cellular redox state. The treatment of cells with 6-S resulted in a burst of intracellular ROS, as evidenced by DCFH-DA staining (Figure 4A) and flow cytometric analysis (Figure 4B). However, almost no ROS were detected in Etc1 cells, further confirming the specificity of 6-S in tumor cells. It is well known that the balance between thiols and disulfides plays a key role in the intracellular redox state. DTNB titration was then used to determine the levels of cellular thiols. After cells were treated with 6-S, the total thiol level decreased (Figure 4C) in HeLa cells, especially at high concentrations of 6-S, and no change was observed in Ect1 cells. Based on previous studies [25,28,29], TrxR inhibitors that contain Michael acceptors can modify the enzyme and eventually convert TrxR into a source of ROS. Therefore, we hypothesized that 6-S might have a similar effect. As shown in Figure 4D, with the consumption of NADPH, 6-S-modified TrxR1 showed ongoing cytochrome C reduction activity, which was subsequently inhibited by SOD, indicating the generation of superoxide anions in this process. As 6-S drastically perturbed the intracellular redox balance and changed the intracellular redox environment, we further explored whether 6-S had a synergistic effect on hydrogen-peroxide (H_2_O_2_)-induced cell death. As shown in Figure 4E, 6-S (10 μM or 20 μM) or H_2_O_2_ (200 μM) did not cause apparent cell death, but their combination significantly enhanced cell death. These findings revealed that 6-S severely destroyed the redox balance and caused oxidative-stress-induced cell death.

### 2.5. Involvement of TrxR for the Cellular Action of 6-S

Next, we established two cell lines: control cells transfected with non-targeting shRNA (HeLa-shNT cells) and cells transfected with shRNA specifically targeting TrxR1 (HeLa-shTrxR1 cells). To determine the transfection efficiency of knockdown, we measured protein expression in these two cell lines. As illustrated in Figure 5A, the expression of TrxR1 in HeLa-shTrxR1 cells was lower than that in HeLa-shNT cells. Next, we determined the cytotoxicity of 6-S in the HeLa-shNT and HeLa-shTrxR1 cells. 6-Shogaol exerted higher cytotoxicity toward HeLa-shTrxR1 cells (Figure 5B). As the Trx system plays a vital role in maintaining the cellular redox balance, we further explored whether TrxR1 was a target for 6-S to induce ROS. Here, we compared ROS generation in HeLa-shNT and HeLa-shTrxR1 cells. After treatment with 6-S for 5 h, a significant difference was observed, especially when a higher concentration of 6-S was used (Figure 5C). R.F.I. from individual cells was measured using ImageJ (Figure 5D). Taken together, these results strongly support that TrxR1 is closely linked to the biological action of 6-S in HeLa cells.

### 2.6. Function of NAC and GSH in 6-S-Induced Cell Death

Glutathione (GSH) is the key part of the GSH system, which is another central thiol redox system in cells and serves as a backup for the Trx system [30]. N-acetylcysteine (NAC) is a known antioxidant and a precursor of GSH, and buthionine sulfoximine (BSO) is a synthesis inhibitor of GSH. In this study, we evaluated the effects of NAC and BSO on 6-S-induced cell death. The pre-incubation of HeLa cells with NAC alleviated 6-S-induced cytotoxicity (Figure 6A). Moreover, using a higher concentration of NAC almost terminated the cytotoxicity of 6-S. BSO is a specific inhibitor widely used to inhibit GSH synthesis in many biological experiments. Under our experimental conditions, the pre-incubation of HeLa cells with 50 μM BSO for 24 h decreased the total GSH level to 20% of the control. After pretreatment of cells with BSO, the cytotoxicity of 6-S was remarkably enhanced (Figure 6B). The addition of NAC reduced the cytotoxicity of 6-S, and GSH depletion increased the cell sensitivity to 6-S, supporting the involvement of the Trx system in the biological action of 6-S.

### 2.7. Induction of Apoptosis by 6-S in HeLa Cells

The results from the double staining (Annexin V-FITC/PI) showed that treatment of HeLa cells with 6-S (50 μM) for 24 h led to the accumulation of apoptotic cells, accounting for over 70% of total cells (Figure 7A). These data suggest that apoptosis is the primary mechanism of the cells’ death. Moreover, cells treated with 6-S displayed highly fluorescent and condensed nuclei (Figure 7B), which are morphological characteristics of apoptosis. In addition, caspase-3 is a biomarker of the apoptotic machinery, and its activation of caspase-3 is a significant step in apoptosis [31]. As shown in Figure 7C, 6-S remarkably increased caspase-3 activity in HeLa cells and almost no change was observed in Ect1 cells. Taken together, these data revealed that 6-S induced cell death mainly through activation of the intrinsic apoptotic pathway.

## 3. Discussion

Many recent studies have focused on the development of new cancer treatment strategies. The discovery of natural compounds targeting TrxR may represent a unique chemotherapeutic option [32]. TrxR has received increased interest because it has been demonstrated to be a signaling intermediate in addition to its inherent antioxidant action. 6-Shogaol, the bioactive component of ginger, exerts significant anticancer activity. Nevertheless, to date, no report has described the mechanism underlying the effect of 6-S on the Trx system. In this study, our findings demonstrated that 6-S induces apoptosis in HeLa cells by targeting TrxR.

TrxR is overexpressed in numerous aggressive cancers [33], which are associated with medication resistance. Additionally, TrxR levels are essential for determining the clinical outcomes in lymphoblastic leukemia [34] and other cancers [35]. This study shows that 6-S effectively suppresses recombinant TrxR1 in a cell-free system. We evaluated the specificity of 6-S by comparing its inhibitory activity against U498C TrxR1, WT TrxR1, and GR, a TrxR homolog [36]. 6-Shogaol selectively inhibited WT TrxR1 in a concentration-dependent manner. Mutation of a 498 Sec to Cys (U498C TrxR1) substantially diminished the inhibition of TrxR1 to 6-S, suggesting that the 498 Sec residue in TrxR1 is an important site of action for 6-S. The weak inhibition of GR by 6-S initially indicated that 6-S might not disrupt the GSH system. To further clarify the reaction site of TrxR1, we used molecular docking to explore the interaction between 498 Sec and 6-S. TrxR1 catalytic 498 Sec was shown to be an elective target, as assessed by molecular docking analyses, which indicated that 6-S could selectively bind TrxR1 by forming a covalent bond with 498 Sec. Next, we evaluated the cytotoxicity of 6-S in HeLa cells and Ect1 cell lines. We found that the cytotoxicity of 6-S was higher in HeLa cells than that in Ect1 cells. Furthermore, 6-S treatment barely affected TrxR activity and thiol levels in Ect1 cells. The specificity of 6-S toward cancer cells demonstrated that 6-S could be a prospective candidate for anticancer drugs.

Cancer cells are known to have high ROS levels due to a malfunction in the respiratory chain and dysregulation of redox balance [37]. A relatively temperature-upgraded ROS in cancer cells can promote tumor survival and growth by activating several enzymes, including cyclin D, extracellular signal-regulated kinase (ERK), and mitogen-activated protein kinase (MAPK) [38]. High levels of ROS can trigger cell death via various mechanisms, including ASK1 activation, stimulation of p53 expression, transfer of cytochrome C from mitochondria to the cytosol, or oxidation of nucleotides in the nucleotide pool [39,40]. Numerous studies have indicated that cellular redox pathways are potentially effective anticancer targets [4,37]. The present study demonstrated that 6-S inhibited TrxR activity in whole-cell lysates, subsequently increasing ROS generation and eventually interrupting cellular redox balance. Modifying TrxR1 with 6-S not only converted this enzyme into NADPH oxidase but also suppressed the reduction of Trx1. The accumulation of NADPH oxidase and oxidized Trx can cause a burst of ROS and affect many Trx-dependent pathways. Furthermore, 6-S combined with H_2_O_2_ had a synergistic effect that induced apoptosis, indicating that oxidative stress plays a key role in the cytotoxicity of 6-S.

The Trx and GSH systems are the two main constituents of the cellular antioxidant network, with overlapping functions in maintaining cellular redox homeostasis. In this study, 6-S treatment significantly inhibited TrxR activity, followed by a block in the Trx system. However, NAC neutralized this effect by restoring the redox balance that was damaged by the collapse of the Trx system. Moreover, the addition of BSO depleted the cellular GSH and enhanced the cytotoxicity of 6-S. These two results indicate that the Trx system is involved in the actions of 6-S and that GSH functions as a backup of the Trx system. TrxRs and GRs are flavoproteins of the same enzyme family; however, only TrxRs are selenoproteins. The active site of TrxRs contains Sec residues, which enable TrxRs to catalyze the reduction of a broad range of substrates. However, TrxRs are easily attacked by electrophilic compounds. Consistent with previous studies [28,29], compounds with *α,β*-unsaturated aldehyde/ketone groups were likely to employ a general inactivation mechanism based on Michael addition between the Sec residues on TrxR and *α,β*-unsaturated aldehyde/ketone on 6-S. Based on the structural analysis of 6-S and the findings shown in Figure 2B,D, we speculated that 6-S suppressed TrxR in HeLa cells via this inactivation mechanism. Intriguingly, 6-S treatment did not affect cellular GSH levels and GR activity, indicating that 6-S selectively disturbs the Trx system but not the GSH system.

Resistance to chemotherapy is a significant hurdle in cancer management. Cancer cells often shift their cellular redox balance to resist different therapeutic approaches. Cancer cells possess a much stronger ability to scavenge ROS to counteract anticancer treatment than normal cells. Since TrxR plays an essential role in maintaining redox balance, targeting TrxR is a potential anticancer strategy. Our data indicated that 6-S promoted oxidative-stress-mediated apoptosis in HeLa cells by inhibiting TrxR. Moreover, they also shed light on the mechanisms underlying its anticancer effect. In this study, HeLa cells were used as a model cell line. Our results provide a strong foundation for elucidating the anticancer effects of 6-S in other cancer cell lines and animal models of different cancers.

## 4. Materials and Methods

### 4.1. Chemicals and Enzymes

Dulbecco’s modified Eagle’s medium (DMEM), bovine insulin, dimethyl sulfoxide (DMSO), Hoechst 33342, N-acetyl-Asp-Glu-Val-Asp-p-nitroanilide (AcDEVD-pNA), yeast glutathione reductase (GR), *N*-acetyl-L-cysteine (NAC), L-buthionine-(S,R)-sulfoximine (BSO), reduced and oxidized glutathione (GSH and GSSG), 2′,7′-dichlorofluorescein diacetate (DCFH-DA), 3-[(3-cholamidopropyl) dimethylamino]-1-propane sulfonate (CHAPS), and superoxide dismutase (SOD) were purchased from Sigma-Aldrich (St. Louis, MO, USA). Penicillin, streptomycin, and 3-(4,5-dimethylthiazol-2-yl)-2,5-diphenyltetrazolium bromide (MTT) were purchased from Amresco (Solon, OH, USA). Cytochrome C was obtained from Sangon Biotech (Shanghai, China). NADPH was obtained from Roche (Mannheim, Germany). Fetal bovine serum (FBS) was purchased from Sijiqing (Hangzhou, China). Anti-Trx1 and anti-TrxR1 antibodies were purchased from Santa Cruz Biotechnology (Dallas, TX, USA). The anti-actin antibody, sodium orthovanadate (Na_3_VO_4_), 2,3-dimercapto-1-propanesulfonic acid (DMPS), phenylmethylsulfonyl fluoride, and bovine serum albumin were obtained from Beyotime (Nantong, China). 5,5′-Dithiobis-2-nitrobenzoic acid (DTNB) and tris(2-carboxyethyl) phosphine hydrochloride (TCEP) were obtained from J&K Scientific (Beijing, China). Recombinant U498C TrxR1 (Sec-Cys), recombinant *Escherichia coli* Trx, and rat TrxR1 proteins were prepared as described in a previous publication [41]. Coomassie-stained sodium dodecyl sulfate (SDS)-polyacrylamide gel electrophoresis (PAGE) was employed to determine the purity of the proteins, and the DTNB assay was used to measure the activity of recombinant TrxR1, which was half of the activity of native TrxR1. In every experiment, the final DMSO concentration was no greater than 0.1% (*v*/*v*). All other reagents used were of analytical grade. 6-S, 6-G, and 6-DG were obtained from Pufei de Biotechnology (Chengdu, China). A 200 mM stock solution of 6-S, 6-G, and 6-DG dissolved in DMSO was stored at −20 °C.

### 4.2. Cell Culture

HeLa cell line and Ect1 cell line were purchased from the Shanghai Institute of Biochemistry and Cell Biology at the Chinese Academy of Sciences. Cells were cultured in DMEM that was supplemented with 10% FBS, 100 units/mL penicillin/streptomycin, and 2 mM glutamine and placed in a humidified 5% CO_2_ incubator at 37 °C.

### 4.3. MTT Assay

Cell viability was measured using an MTT assay. Cells were seeded at a specific density in 96-well plates. After incubation with 6-S or other reagents for the indicated time, cells were sequentially incubated with 10 μL MTT for an additional 4 h. Formazan was dissolved in 100 μL extraction buffer containing 0.1% HCl, 5% isobutanol, and 10% SDS. After 12 h, absorbance at 570 nm was recorded using a microplate reader (Thermo Scientific Multiskan GO, Vantaa, Finland). Cell viability was calculated as the percentage of absorbance of treated and control cells.

### 4.4. TrxR Activity Assays In Vitro

Purified TrxR activity was measured using the DTNB assay and endpoint insulin reduction assay [42,43].

#### 4.4.1. DTNB Assay

The assay master mixture containing 2 mM DTNB and 200 μM NADPH was prepared in Tris-ethylenediaminetetraacetic acid (EDTA) (TE) buffer (50 mM Tris-HCl, pH 7.5, 1 mM EDTA). After different concentrations of 6-S were incubated with U498C TrxR (700 nM) or NADPH-reduced TrxR (80 nM) in a volume of 50 μL for the indicated times at room temperature, a 50 μL master mixture was added. The change in absorbance (412 nm) during the first 3 min was immediately recorded. The inhibitory effect of 6-S was expressed as a percentage of the treated group relative to the control group.

#### 4.4.2. Endpoint Insulin Reduction Assay

The purified TrxR activity was measured using an endpoint insulin reduction assay [28]. The assay master mixture, containing 0.32 mM insulin, 660 μM NADPH, and 15 μM *E. coli* Trx, was prepared in TE buffer (50 mM Tris-HCl, pH 7.5, 1 mM EDTA), and DTNB was dissolved in 6 M guanidine hydrochloride (pH 8.0) at a concentration of 1 mM. After different concentrations of 6-S were made to react with NADPH-reduced TrxR (80 nM) for 2 h in a volume of 50 μL at room temperature, the solution was incubated with a 50 μL master mixture for another 0.5 h. Then, 100 μL DTNB (1 mM) was added to terminate the reaction. The absorbance (412 nm) was monitored, and TrxR activity was calculated as the percentage of the treated group relative to the control group.

### 4.5. GR Assay

An aliquot of the 100 μL mixture containing 6-S and NADPH-reduced GR (0.25 unit/mL) was pre-incubated for 2 h at room temperature. The reaction was initiated by adding an aliquot of a 50 μL mixture containing NADPH (400 μM) and GSSG (1 mM). A decrease in the absorbance (340 nm) during the first 3 min was recorded [44]. GR activity is expressed as a percentage of the treated group relative to the control group.

### 4.6. Molecular Docking

This work applied the Schrödinger software for molecular docking to depict the potential association of TrxR1 with 6-S according to a previously described method [28]. The rat TrxR1 structure was obtained from the Protein Data Bank (PDB code 3EAN, Chains A and B) and subsequently incorporated into the protein preparation wizard module. Usually, the reactive residue Sec498 of Chain A is chosen as the centroid of the docking pocket and a Michael addition-related reactive residue. The default settings were used for the docking simulation.

### 4.7. Determination of Cellular TrxR Activity

The endpoint insulin reduction assay was used to detect cellular TrxR activity. After treatment with 6-S, cells were harvested and lysed in radioimmunoprecipitation assay (RIPA) buffer (50 mM Tris-HCl, pH 7.5, 0.5% deoxycholate, 0.1% SDS, 150 mM NaCl, 1 mM Na_3_VO_4_, 1 mM phenylmethylsulfonyl fluoride, 2 mM EDTA, and 1% Triton X-100). Bradford assay was used to measure the total protein content in the cell lysate. Cellular TrxR activity was detected according to our published protocols [43].

### 4.8. Imaging Cellular TrxR Activity Using TRFS-Green

TRFS-green, a specific dye synthesized by Fang’s laboratory [45], was used to detect the activity of cellular TrxR. Cells (2 × 10^5^ cells/well) were seeded in 12-well plates and cultured for one day. The cells were treated with different concentrations of 6-S for 20 h. TRFS-green (10 μM) was added as a cellular marker, and cells were incubated for 4 h in the dark at 37 °C. An inverted fluorescence microscope (Leica DMI4000 Microsystems GmbH, Wetzlar, Germany) was used to capture the fluorescence images. Green fluorescence in the cells indicated the relative activity of TrxR.

### 4.9. Intracellular ROS Detection

Cells (2 × 10^5^ cells/well) were seeded in 12-well plates and cultured for one day. Cells were then treated with 6-S for 5 h, DCFH-DA (10 μM) in fresh FBS-free medium was added, and the incubation was continued for 15 min at 37 °C in the dark. Images were captured using an inverted fluorescence microscope (Leica DMI4000 Microsystems GmbH, Germany).

Intracellular ROS levels were determined using flow cytometry. Briefly, after treatment with 6-S, cells were collected and suspended in PBS. DCFH-DA (10 μM) was then added, and incubation was continued for 15 min at 37 °C in the dark. After washing with ice-cold PBS twice, the cells were immediately analyzed using a flow cytometer (BD Biosciences, San Jose, CA, USA) at an emission wavelength of 525 nm. The relative fluorescence intensity in cells was quantified using ImageJ.

### 4.10. NADPH Oxidase Assay and Cytochrome C Reduction Assay

After 6-S (500 μM) was made to react with NADPH-reduced TrxR (1.45 μM) in TE buffer at room temperature for 2 h, the DTNB reduction assay was employed to monitor the remaining enzyme activity, which was less than 10% of the control. Unreacted 6-S was then separated from the reaction solution using a Sephadex G-25 desalting column (GE Healthcare Life Sciences, Chicago, IL, USA). To detect NADPH oxidase activity, NADPH (200 μM) was added to 44 μL of the modified enzyme in a final volume of 300 μL. Absorbance at 340 nm was recorded to calculate the oxidation of NADPH (ε = 6200 M^−1^ cm^−1^). Next, a cytochrome C reduction assay was performed to determine the production of superoxide anions. An amount of 34 μL cytochrome C (0.82 mM) was added to the reaction solution, and the absorbance spectrum (500–650 nm) was recorded. At the indicated time points, 300 U of SOD was added to scavenge the superoxide anions. The increase in absorbance (550 nm) between conditions with and without SOD was analyzed to evaluate the production of superoxide anion (ε = 21,000 M^−1^ cm^−1^) [25].

### 4.11. Assessment of Intracellular Thiol Levels

Total thiol levels were measured according to the DTNB titration method described by Ellman et al. [46]. Cells were treated with 6-S in a six-well plate for 24 h. Then, the cells were harvested and lysed in RIPA buffer. Bradford assay was used to measure the total protein content of the cell lysates. Intracellular thiol levels were assessed as previously described in our published protocol [28].

### 4.12. Determination of the Trx Redox State

According to our previously published procedures [47,48], the changes between reduced Trx and oxidized Trx were measured using phenylarsine oxide (PAO)-sepharose. Cells were treated with 6-S for 24 h, after which protein was extracted from the cells. The Bradford assay was used to quantify the protein content. The untreated cell lysate incubated with diamide (5 mM) or TCEP (5 mM) was used as the fully oxidized or reduced control. The mixture containing the samples and PAO-sepharose was placed on a rotating shaker for 30 min at room temperature. The oxidized Trx in the supernatant was separated from the reduced Trx in PAO-sepharose, and the reduced Trx was eluted from the sepharose using DMPS (20 mM). All recovered samples were analyzed by western blotting.

### 4.13. Annexin V/Propidium Iodide (PI) Staining

After treatment with 6-S for 24 h, cells were collected and washed with ice-cold PBS. The cells were then resuspended in a 500 μL solution containing PI and Annexin V/FITC (Zoman Biotech, Beijing, China). The double-labeled cells were monitored using a flow cytometer (BD Biosciences). The results were summarized using CellQuest software (BD Biosciences).

### 4.14. Hoechst 33342 Staining

HeLa cells (2 × 10^5^ cells/well) were seeded in 12-well plates and cultured for one day. After the cells were treated with 6-S for 5 h, Hoechst 33342 (5 μg/mL) was added, and incubation was continued for 20 min at 37 °C in the dark. Images were captured using an inverted fluorescence microscope (Leica DMI4000 Microsystems GmbH, Germany).

### 4.15. Measurement of Caspase-3 Activity

An assay mixture (50 mM HEPES, 5% glycerol, 2 mM EDTA, 0.1% CHAPS, 10 mM DTT, 0.2 mM Ac-DEVD-pNA, pH 7.5) was prepared for the follow-up experiment. After treatment with 6-S for 24 h, cells were harvested and lysed in RIPA buffer. Bradford assay was used to measure the total protein content in the cell lysate. Then, a 100 μL solution containing 30 μg of protein and assay mixture was incubated at 37 °C for 2 h. The absorbance (405 nm) was recorded, and caspase-3 activity was calculated as the percentage of the treated group relative to the control group.

### 4.16. Statistics

Values are reported as mean ± standard error of three independent experiments. The Student’s *t*-test was used to assess statistical differences between the two groups. Multiple comparisons were performed using a one-way analysis of variance (ANOVA). Statistical significance was set at *p* < 0.05.

## 5. Conclusions

We have shown that 6-S treatment inhibits TrxR in HeLa cells and schematically presented the molecular mechanism of 6-S (Figure 8). The significant findings of this study are as follows: (1) 6-S exhibits higher cytotoxicity compared with 6-G and 6-DG, and also selectively induces death in HeLa cells as compared with Ect1 cells; (2) 6-S not only inhibits purified TrxR but also intracellular TrxR in HeLa cells, and the inactivation mechanism is possibly achieved by the irreversible covalent addition between the Sec residues on TrxR and α,β-unsaturated ketone on 6-S; (3) 6-S promotes the ratio of oxidized Trx to reduced Trx and further destroys the Trx system; (4) knockdown of TrxR enhances cytotoxicity and ROS production in HeLa cells; and (5) 6-S induces oxidative-stress-mediated apoptosis in HeLa cells. This study provides new insights into the specific mechanisms employed by 6-S, suggesting that 6-S is a promising candidate for further development as a therapeutic anticancer agent.

## Figures and Tables

**Figure 1 ijms-24-04966-f001:**
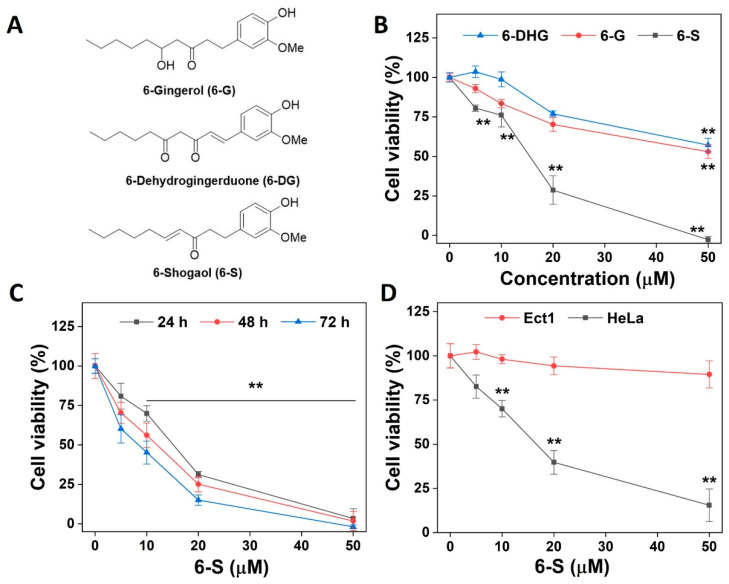
Induction of cell death by 6-S. (**A**) The chemical structure of 6-G, 6-DG, and 6-S. (**B**) Effect of 6-G, 6-DG, or 6-S on the cytotoxicity of HeLa cells. The cells (5 × 10^3^/well) were treated with the indicated concentrations of compounds for 24 h, and the cell viability was determined by the MTT assay. (**C**) Sensitivity of HeLa cells to 6-S. The cells (5 × 10^3^/well) were treated with 6-S for the indicated time, and the viability was determined using the MTT assay. (**D**) Cytotoxicity of 6-S in HeLa and Etc1 cells. The cells (1 × 10^4^) were treated with 6-S for 48 h, and the viability was determined using the MTT assay. Data are expressed as the mean ± standard error of means of three experiments. **, *p* < 0.01 vs. the control group.

**Figure 2 ijms-24-04966-f002:**
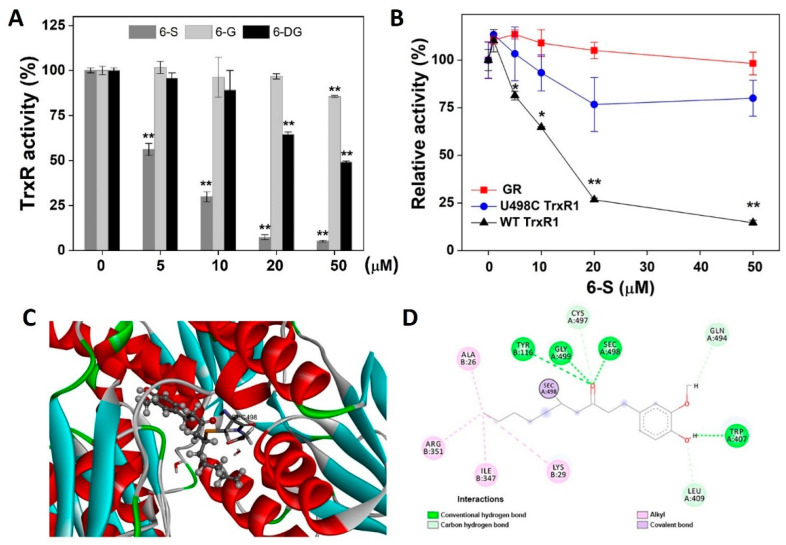
Inhibition of TrxR activity by 6-S. (**A**) Dose-dependent inhibition of TrxR by 6-S, 6-G, and 6-DG. The activity of TrxR was determined using the endpoint insulin reduction assay. (**B**) Inhibition of WT TrxR, U498C TrxR, and GR by 6-S. WT TrxR1, U498C TrxR1, and GR were incubated with the indicated concentrations of 6-S for 2 h at room temperature, and the enzyme activities were determined. (**C**) The surface representation of the reactive site of 6-S on rat TrxR1. (**D**) Covalent binding mode of the rat TrxR1 and 6-S. Data are expressed as the mean ± standard error from three experiments. *, *p* < 0.05, and **, *p* < 0.01 vs. the control group.

**Figure 3 ijms-24-04966-f003:**
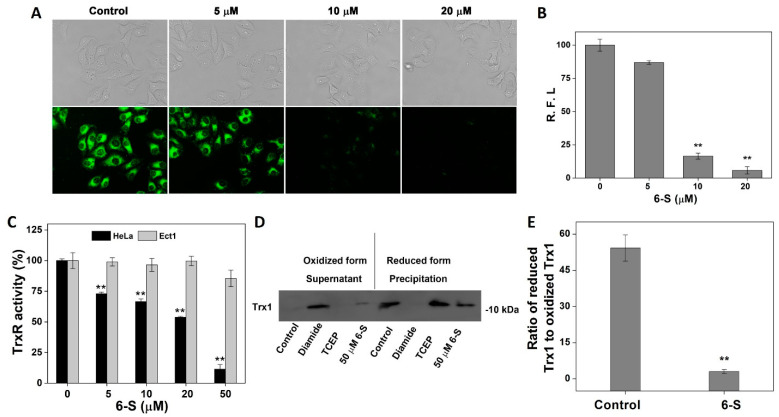
Influence of 6-S on Trx system. (**A**) Detection of TrxR activity using TRFS-green in HeLa cells. The phase-contrast (top panel) and fluorescence (bottom panel) microscopy images were captured using an inverted fluorescence microscope. Scale bars: 20 μm. (**B**) Quantification of R.F.I. in individual cells by ImageJ. (**C**) Inhibition of TrxR activity in HeLa cells and Etc1 cells. After the cells were treated with the indicated concentrations of 6-S for 24 h, the cellular TrxR activity was measured using the endpoint insulin reduction assay. (**D**) Determination of the Trx redox state. HeLa cells were treated with vehicle or 6-S (0 μM or 50 μM) for 24 h, and the Trx1 redox states were measured. O: oxidized form, R: reduced form, S: Supernatant, P: Precipitation. (**E**) Quantitative analysis based on the ratio of reduced Trx1 to oxidized Trx1 by measuring the band density using IPP. Data are expressed as the mean ± standard error from three experiments. **, *p* < 0.01 vs. the control group.

**Figure 4 ijms-24-04966-f004:**
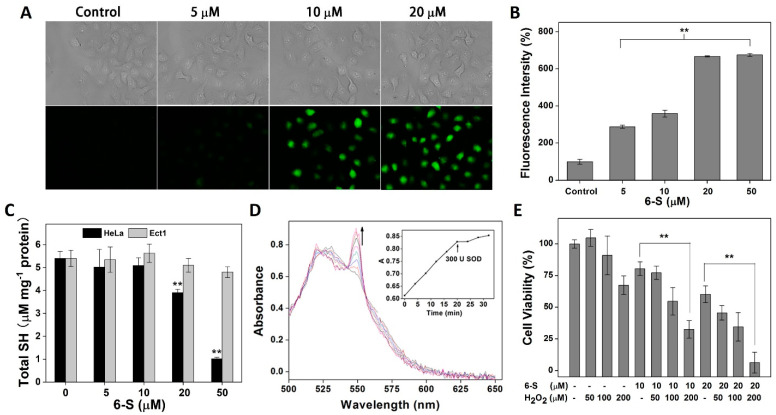
Induction of oxidative stress by 6-S. (**A**) The ROS level detected by DCFH-DA staining in HeLa cells. Phase-contrast (top) and fluorescence (bottom) images were acquired by inverted fluorescence microscopy. Scale bars: 20 μm. (**B**) Detection of ROS with DCFH-DA in HeLa cells by Flow Cytometry. (**C**) Consumption of intracellular thiols by 6-S in HeLa cells and Etc1 cells. Free thiols in the total cell extract were quantified by DTNB titration. (**D**) Induction of superoxide anion production by 6-S-modified TrxR1. Superoxide anion generation was monitored by cytochrome C reduction assay. The inset shows a change in absorbance at 550 nm after adding cytochrome C and SOD. SOD was added at the time point indicated by the arrow in the inset. (**E**) Enhancement of H_2_O_2_ cytotoxicity by 6-S. HeLa cells (1 × 10^4^) were seeded on 96-well plates and allowed to attach for 24 h. Then, cells were pretreated with 6-S (10 or 20 μM) for 4 h. After removing 6-S, the cells were further incubated with H_2_O_2_ (50, 100, or 200 μM) for 24 h. Cell viability was determined by MTT assay. Data are expressed as means ± SE of three experiments. **, *p* < 0.01, vs. the control group.

**Figure 5 ijms-24-04966-f005:**
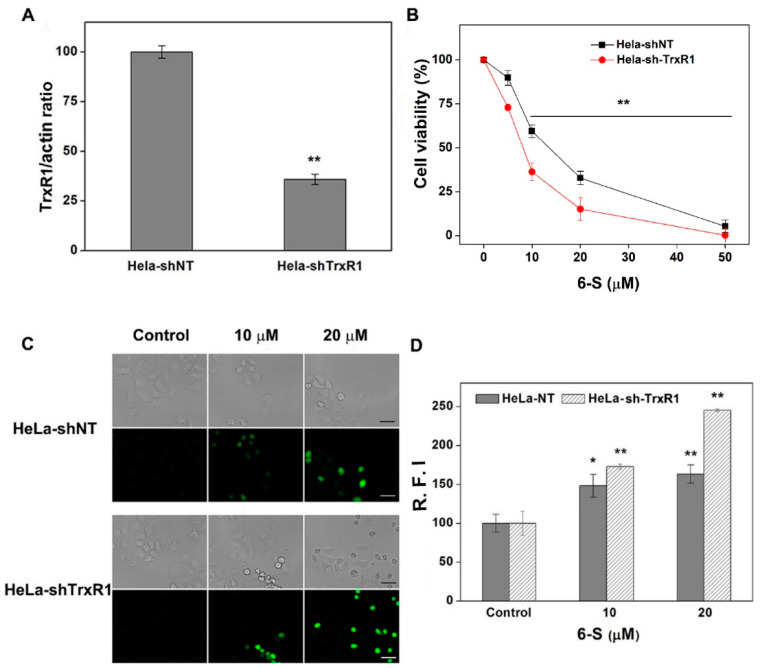
Involvement of TrxR for the cellular action of 6-S. (**A**) Quantification of TrxR expression in HeLa-shNT and HeLa-shTrxR1 cells by IPP. Cell extracts were prepared and analyzed by western blotting with an antibody against TrxR1 and actin (inset). (**B**) Cytotoxic effects of 6-S on HeLa-shNT and HeLa-shTrxR1 cells. The cells (5 × 10^3^) were treated with the indicated concentrations of 6-S for 48 h, and the cell viability was determined by the MTT assay. (**C**) The ROS levels in HeLa-shNT and HeLa-shTrxR1 after 6-S treatment. Scale bars: 20 μm. (**D**) Quantification of R.F.I. in HeLa-shNT and HeLa-shTrxR1 cells by ImageJ. Data are expressed as means ± SE of three experiments. *, *p* < 0.05, and **, *p* < 0.01, vs. the control group.

**Figure 6 ijms-24-04966-f006:**
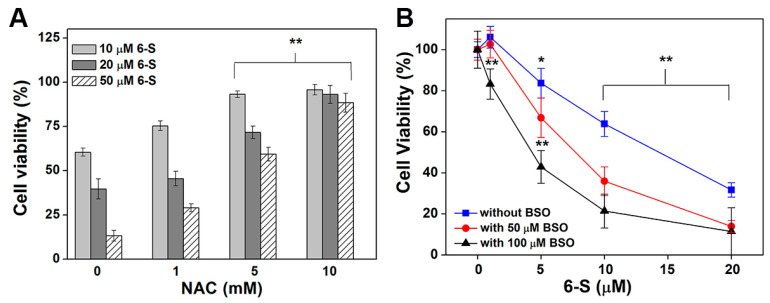
Function of NAC and GSH in 6-S-induced cell death. (**A**) Protection of the cells by NAC. HeLa cells (1 × 10^4^) were seeded in 96-well plates. After 24 h, the cells were incubated with the indicated concentrations of NAC and 6-S (10, 20, or 50 μM) for another 24 h. Cell viability was determined by the MTT assay. (**B**) Enhancement of cytotoxicity by BSO. HeLa cells (5 × 10^3^) were treated with 50 μM or 100 μM BSO for 24 h and followed by 6-S treatment for an additional 72 h. Cell viability was determined by the MTT assay. *, *p* < 0.05, and **, *p* < 0.01, vs. the control group.

**Figure 7 ijms-24-04966-f007:**
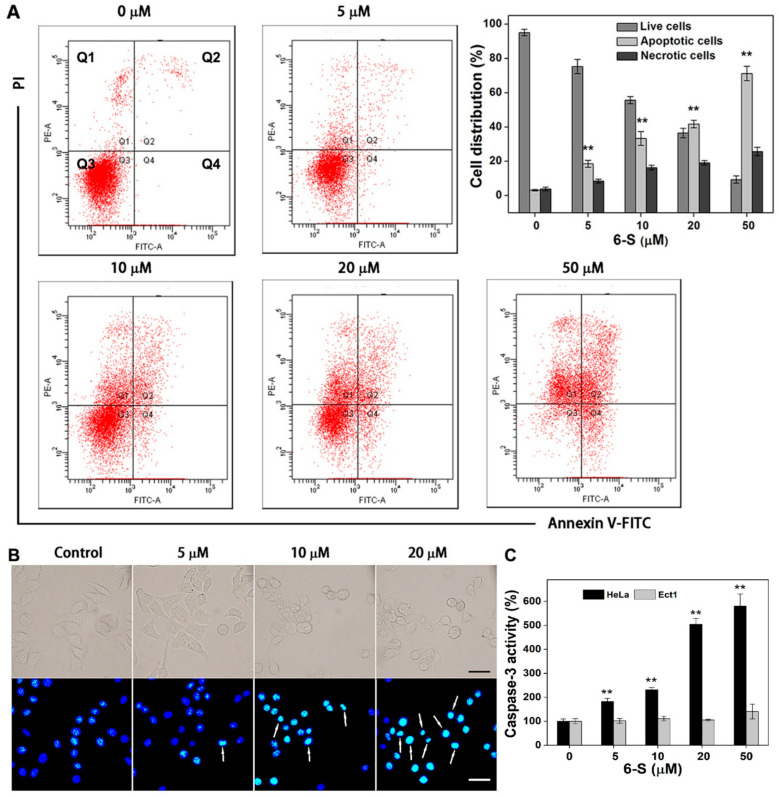
Induction of apoptosis by 6-S in HeLa cells. (**A**) Analysis of apoptosis by Annexin V-FITC/PI double-staining assay. The representative FACS analysis scattergrams of annexin V-FITC/PI staining are shown. The cells show four different cell populations, marked as follows: double-negative (unstained) cells showing the live cell population (lower left, Q3); Annexin V-FITC-positive- and PI-negative-stained cells showing early apoptosis (lower right, Q4); Annexin V/PI-double-stained cells showing late apoptosis (upper right, Q2); and finally, PI-positive- and Annexin V-negative-stained cells showing necrotic cells (upper left, Q1). The quantifications of live cells (Q3), apoptotic cells (Q2 and Q4), and necrotic cells (Q1) are illustrated in the upper-right corner. (**B**) Monitoring the apoptosis by nuclear condensation. The Hoechst 33342 staining showed typical apoptotic morphology changes after the 6-S treatment. Phase-contrast (top) and fluorescence (bottom) images were acquired by inverted fluorescence microscopy. Scale bars: 20 μm. (**C**) Activation of caspase-3 by 6-S in Hela cells and Etc1 cells. Colorimetric assay determined the caspase-3 activity in cell extracts. ** *p* < 0.01, vs. the control group.

**Figure 8 ijms-24-04966-f008:**
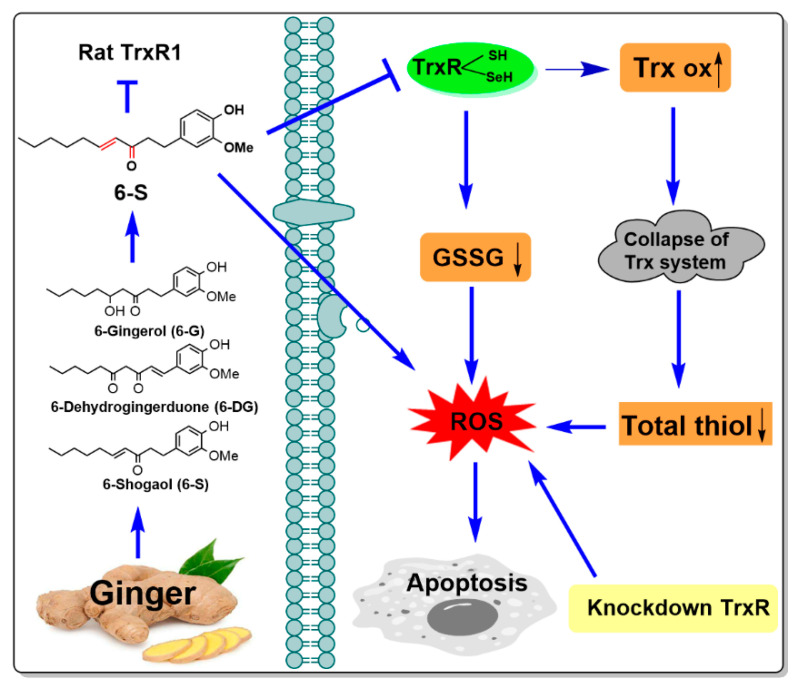
Targeting the TrxR by 6-S induces oxidative stress in HeLa cells.

## Data Availability

The data presented in this study are available upon request from the corresponding author.

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
