# Peer review of "6-Shogaol as a Novel Thioredoxin Reductase Inhibitor Induces Oxidative-Stress-Mediated Apoptosis in HeLa Cells"

_ijms, 2023, doi:10.3390/ijms24054966_

Round 1
Reviewer 1 Report
The submitted article by Peng et al reports that a novel thioredoxin reductase inhibitor, 6-Shogaol (6-S), induces oxidative stress in HeLa cells resulting in their apoptosis. They further show that the molecular mechanism of 6-S-mediated apoptosis involves TrxR inhibition followed by induction of ROS production resulting in cytotoxicity and suggest that 6-S could be a candidate for further development as a novel anticancer therapeutic.
In general, I have few problems with the data in any of the figures – the results and conclusions seem to be based on experiments that have been carefully carried out and sensibly interpreted.
However, the crux of the manuscript is the authors’ view that 6-S discriminates between Hela cells (as a cell model of cancer) and “normal cells” (line 490) and I have a major problem with their choice of “normal cells” right from the beginning of their results.
HEK293T cells are a T antigen transformed human embryonic kidney cell line – and it is entirely possible that, for unknown reasons, are more resistant to 6-S than e.g. a more biologically relevant “normal cells”.
Similarly, L02 cells are unfortunately not “normal cells” either. Indeed, L02 cells are known to be a problematic cell line - originally thought to be from normal fetal liver, they have recently been shown to be a HeLa derivative (see FASEB J. 2015;29:4268–72 and DOI: 10.1002/hep.32730).
Consequently, the authors use of L02 cells as normal cells may be problematic. It is, therefore, important to recapitulate some of the experiments (e.g. figure 1D) with a more biologically relevant normal cell (e.g. ECT1 cells could be used). Additionally, I am uncertain why normal control cells were not also analyzed in e.g. figures 3 and 4 and particularly figure 7.
Minor comments:
(i) The author list appears incomplete.
Author Response
Response to Reviewer 1 Comments
1. The submitted article by Peng et al reports that a novel thioredoxin reductase inhibitor, 6-Shogaol (6-S), induces oxidative stress in HeLa cells resulting in their apoptosis. They further show that the molecular mechanism of 6-S-mediated apoptosis involves TrxR inhibition followed by induction of ROS production resulting in cytotoxicity and suggest that 6-S could be a candidate for further development as a novel anticancer therapeutic. In general, I have few problems with the data in any of the figures – the results and conclusions seem to be based on experiments that have been carefully carried out and sensibly interpreted.
Response 1: We thank you for giving us a chance to revise the manuscript and appreciate your positive comments on our work.
2. However, the crux of the manuscript is the authors’ view that 6-S discriminates between Hela cells (as a cell model of cancer) and “normal cells” (line 490) and I have a major problem with their choice of “normal cells” right from the beginning of their results.
HEK293T cells are a T antigen transformed human embryonic kidney cell line – and it is entirely possible that, for unknown reasons, are more resistant to 6-S than e.g. a more biologically relevant “normal cells”.
Similarly, L02 cells are unfortunately not “normal cells” either. Indeed, L02 cells are known to be a problematic cell line - originally thought to be from normal fetal liver, they have recently been shown to be a HeLa derivative (see FASEB J. 2015;29:4268–72 and DOI: 10.1002/hep.32730).
Consequently, the authors use of L02 cells as normal cells may be problematic. It is, therefore, important to recapitulate some of the experiments (e.g. figure 1D) with a more biologically relevant normal cell (e.g. ECT1 cells could be used). Additionally, I am uncertain why normal control cells were not also analyzed in e.g. figures 3 and 4 and particularly figure 7.
Response 2: This comment is quite relevant and important. After reading the ref (FASEB J. 2015; 29:4268-72 and DOI: 10.1002/hep.32730) and other related refs, we have realized that these two normal cell lines we used in our manuscript are problematic and unreasonable. So, here we’ve removed this result (Fig 1D) and related words in whole manuscript.
Minor comments:
(i) The author list appears incomplete.
Response: We are deeply sorry for this mistake. We’ve corrected it.
Reviewer 2 Report
Zhang et al. reported a study about 6-Shogaol as a novel thioredoxin reductase inhibitor that induces oxidative stress-mediated apoptosis in HeLa cells. This research theme is interesting in revealing anticancer compounds from nature. However, several things need to be explained in the manuscript:
1. it is necessary to explain the source of 6S and its derivatives.
2. Inhibition of purified TrxR activity results are reported only by 6-S. How are the results by 6-G and 6-DG? TrxR and 6-G/6-DG docking results should be displayed and discussed.
3. 3.6. The function of NAC and GSH in 6-S-induced cell death was not stated in the study method. Please explain what NAC and GDH stand for and their sources.
4. line 476 describes that resistance to chemotherapy is a significant hurdle in cancer management. Cancer cells often shift their cellular redox balance to resist different therapeutic approaches. It is necessary to explain whether the Hela cells used in the research are Hela cells that are still sensitive or already resistant to chemotherapy.
The recommendation for this manuscript is a minor revision.
Author Response
Response to Reviewer 2 Comments
Zhang et al. reported a study about 6-Shogaol as a novel thioredoxin reductase inhibitor that induces oxidative stress-mediated apoptosis in HeLa cells. This research theme is interesting in revealing anticancer compounds from nature. However, several things need to be explained in the manuscript:
1. It is necessary to explain the source of 6-S and its derivatives.
Response 1: Thank you for your suggestion. We’ve added some new words to explain the source of 6-S and its derivatives (page 4, the green background text).
2. Inhibition of purified TrxR activity results are reported only by 6-S. How are the results by 6-G and 6-DG? TrxR and 6-G/6-DG docking results should be displayed and discussed.
Response 2: Thank you for your question. This comment is quite relevant and important. We’ve added inhibition results of 6-G/6-DG (Fig 2A) and also added some new words to discuss (page 9, the green background text). Besides, we performed the doking simulation between 6-G/6-DG and TrxR, and the results demonstrated that 6-S showed the best docking affinity scores compared with 6-G and 6-DG. As we selected 6-S for the follow-up studies, we just displayed the doking figure of 6-S. But we added some new words to discuss the doking results among 6-S, 6-G and 6-DG (page 9, the green background text).
3. The function of NAC and GSH in 6-S-induced cell death was not stated in the study method. Please explain what NAC and GSH stand for and their sources.
Response 3: Thank you for your question. We’ve added some new words to explain GSH, NAC and BSO (page 11, the green background text). Besides, we’ve also added the source of GSH, NAC and BSO (page 3, the green background text).
4. line 476 describes that resistance to chemotherapy is a significant hurdle in cancer management. Cancer cells often shift their cellular redox balance to resist different therapeutic approaches. It is necessary to explain whether the Hela cells used in the research are Hela cells that are still sensitive or already resistant to chemotherapy.
Response 4: Thank you for your question. The Hela cells used in our research are regular and still sensitive.
The recommendation for this manuscript is a minor revision.
Round 2
Reviewer 1 Report
The authors seem to have simply removed the L02 and HEK293T cell "controls" in response to my original view that these were not "normal" cell types - which they now agree with.
However, they still do not show any analysis of the effects of 6-S on normal cells i.e. there are still no control cells with which to compare killing of HeLa cells. Essentially, there is no evidence that 6-S could be specific for cancer cells.
All my original comments with respect to a need to recapitulate their main observations with a set of control cells still stand - particularly in figures 1, 3 4 and 7.
Author Response
Response to Reviewer 1 Comments
The authors seem to have simply removed the L02 and HEK293T cell "controls" in response to my original view that these were not "normal" cell types - which they now agree with.
However, they still do not show any analysis of the effects of 6-S on normal cells i.e. there are still no control cells with which to compare killing of HeLa cells. Essentially, there is no evidence that 6-S could be specific for cancer cells.
All my original comments with respect to a need to recapitulate their main observations with a set of control cells still stand - particularly in figures 1, 3 4 and 7.
Response: We are deeply sorry for our negligence. Thank you for your patience. We’ve employed Ect1 cells as the normal cell line to perform relative experiments, including Fig 1D (cytotoxicity), Fig 3C (intracellular TrxR activity), Fig 4C (total thiol level) and Fig 7C (caspase-3 activity). 6-S exhibited low cytotoxicity toward Ect1 cells compared with HeLa cells (Fig 1D). Besides, after 6-S treated Ect1 cells for 24 h, no obvious change was observed in intracellular TrxR activity (Fig 3C), total thiol level (Fig 4C) and caspase-3 activity (Fig 7C), highlighting the specificity of 6-S towards cancer cells. We’ve added some words to explain these new data (the green background text).
Round 3
Reviewer 1 Report
The authors appear to have addressed all my comments comprehensively.